# Near–Real Time Burst Location and Sizing in Water Distribution Systems Using Artificial Neural Networks

Miguel Capelo [1], Bruno Brentan [2], Laura Monteiro [1] and Dídia Covas [1,*]

1   CERIS, Instituto Superior Técnico, Universidade de Lisboa, 1049-001 Lisbon, Portugal; migueltavarescapelo2@tecnico.ulisboa.pt (M.C.); laura.monteiro@tecnico.ulisboa.pt (L.M.)
2   Hydraulic Engineering and Water Resources Department, School of Engineering, Federal University of Minas Gerais, Belo Horizonte 31270-901, Brazil; brentan@ehr.ufmg.br
*   Correspondence: didia.covas@tecnico.ulisboa.pt; Tel.: +351-218-418-152

**Abstract:** The current paper proposes a novel methodology for near–real time burst location and sizing in water distribution systems (WDS) by means of Multi–Layer Perceptron (MLP), a class of artificial neural network (ANN). The proposed methodology can be systematized in four steps: (1) construction of the pipe–burst database, (2) problem formulation and ANN architecture definition, (3) ANN training, testing and sensitivity analyses, (4) application based on collected data. A large database needs to be constructed using 24 h pressure–head data collected or numerically generated at different sensor locations during the pipe burst occurrence. The ANN is trained and tested in a real–life network, in Portugal, using artificial data generated by hydraulic extended period simulations. The trained ANN has demonstrated to successfully locate 60–70% of the burst with an accuracy of 100 m and 98% of the burst with an accuracy of 500 m and to determine burst sizes with uncertainties lower than 2 L/s in 90% of tested cases and lower than 0.2 L/s in 70% of the cases. This approach can be used as a daily management tool of water distribution networks (WDN), as long as the ANN is trained with artificial data generated by an accurate and calibrated WDS hydraulic models and/or with reliable pressure–head data collected at different locations of the WDS during the pipe burst occurrence.

**Keywords:** burst location; burst quantification; water distribution networks; Artificial Neural Networks







## 1. Introduction

Water distribution systems (WDS) are one of the most important public infrastructures that provide an essential service to populations: the provision of water in quantity and with adequate pressure and quality. Most WDS in developed countries were constructed decades ago and, currently, have to deal with high water losses and frequent pipe bursts, requiring constant maintenance works and the urgent implementation of rehabilitation plans [1]. The fact is that pipe burst repairs are responsible for the largest part of the operation and maintenance (O&M) budget of water distribution systems [2]. The earlier detected and repaired a burst is, the lower the associated O&M and water losses costs are, as well as a the lower the risk of contaminant intrusion in the system becomes and, as a consequence, the safer the distributed water is [3].

Given the importance of burst detection and location, industry and the scientific community have explored different methods to detect and to locate leaks and ruptures during the last decades. Among the practical water losses control methods and leak detection techniques, acoustic signal analysis, minimum night flow monitoring and water balance calculation in network sectors are the most widely used and successfully implemented by many water utilities [4–7]. Though these methods undoubtedly allow efficient water losses control in WDS, they also require a huge investment of water utilities in training human resources and in the installation of the necessary instrumentation for monitoring the

systems, which is not always possible to many utilities due to lack of human, technological and, above all, financial resources.

Given the constant search for more efficient and lower–cost leak detection methods, many research efforts for leak detection, location and quantification have been done which can be divided in two groups, depending on the type of data used: transient–state data and steady–state data. Transient–state data have been explored by means of different techniques, namely by: simple transient pressure trace analysis to detect the leak signal [8,9], wavelet analysis [10,11], inverse transient analysis [12–15] and frequency analysis [16]. However, the application of transient based techniques in real and large networks requires the installation of many sensors with high accuracy and frequency, which is very expensive and technology demanding, requiring the storage, transmission and processing of a huge amount of data; additionally, networks have many unknown consumptions and several pipe connections and fittings that create multiple reflections that can be misleading in the detection of pipe bursts when using transient–state data. Thus, steady state analysis has also been used for detecting and finding bursts and estimating leakage [17–23].

The last decades have brought the possibility of remote monitoring with the implementation of supervisory control and data acquisition (SCADA) systems to the WDS. Data acquired by SCADA are stored and, if well handled, can be used to improve the O&M of the systems. In this line, algorithms based on machine learning and data mining are useful and widely used for leakage and burst detection in water systems, such as Artificial Neural Networks both using steady–state [24,25] and transient–state data [26,27]. The design of a wireless sensor network joint to a machine learning algorithm to detect and to quantify leaks in water systems has been proposed [28].

The current paper aims at using a data mining and machine learning technique, Artificial Neural Networks (ANN), for near–real time burst location and sizing in water distribution networks. An ANN is trained and tested using pressure–head data generated by numerical simulations of a real water network. A set of 18,696 single bursts with a duration of 4 h, located at one of the 3116 nodes, with six different emitter coefficients has been numerically generated and used to train and to test the ANN. The criteria for assessing the uncertainty of burst correct location and size are: the distance between the true–burst node and the estimated–burst node, represented by X and Y coordinates, and the burst discharge uncertainty given by the different between the real discharge and the one estimated by the ANN. Sensitivity analyses are carried out for different ANN configurations, number of sensors, burst scenarios and the location of the sensors.

The main innovative features of the paper are (i) the use of an ANN for near real–time location of leaks using steady–state data for both leak location and sizing at the end of the 24 h of sensor data collection, an approach that can be replicated with real and artificial data in any WDS; and (ii) the testing and sensitivity analysis of the approach application in a real water distribution network.

## 2. Methodology

### 2.1. General Approach

A novel approach to locate and quantify pipe bursts in water distribution networks, based on the use of ANN, is proposed and described herein. The methodology is a four–step procedure (Figure 1): (1) pipe–burst database construction; (2) problem formulation and ANN architecture definition; (3) ANN training, testing and sensitivity analyses; (4) application based on collected data. These steps are explained in detail in the following sections.

### 2.2. Pipe–Burst Database Construction

The first step is the construction of a wide–ranging and reliable pipe–burst database. This database must contain pressure–head and/or flow–rate data collected at different locations of the water distribution network during a 24 h period, in the days of the occurrence of pipe bursts. It should as well contain the complete characterization of the pipe burst in terms of location, average flowrate, starting time and duration.

In real WDS, the complete characterization of each burst to construct this ideal database is not usually available for two main reasons. Firstly, utilities do not have an integrated data management system articulated with the pipe bursts records. Second, the data may exist, but the installed sensors are either insufficient or not uniformly spread throughout the WDS, and cannot monitor the whole network, or the sensors are enough, but the number of completely recorded cases is not sufficient to represent the burst occurrence in the whole network. Thus, the alternative is to artificially generate these data by using a robust and well–calibrated network model, for instance, developed in EPANET, that can reliably describe the hydraulic behavior of the system during the 24–hour period of analysis.

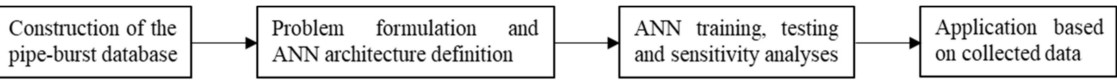

**Figure 1.** Flow chart representing the four–step methodology.

Thus, the pipe–burst database can be composed of a set of records of burst events numerically simulated. Each event data record corresponds to one single pipe burst. Pipe bursts should be simulated at different locations and with different sizes and times of occurrence, randomly generated within certain time intervals. Not all network nodes need to be considered as potential burst locations, since many of them are redundant. As such, a selection of nodes with bursts should be carried out, eliminating, for instance, too–close nodes or nodes located upstream of opened valves when the downstream node is included. Burst sizes should be defined within a reasonable range that can go from detectable leaks (e.g., 0.5 L/s) to not excessively large bursts (e.g., 30 L/s). The occurrence time and duration can be randomly chosen, as long as these are within the simulated 24 h period.

After choosing the potential burst node candidates, the next step is to define the location of the pressure–head and of flow–rate sensors. These should be spread throughout the network. The more sensors exist and the more uniformly distributed these are in the network, the better the quality of collected data is, since the data sets are more representative of the system behavior. This database should contain a significant amount of pipe bursts of the order of thousands: few data are not enough to allow the ANN to learn about the burst location; too many data will delay the ANN training and do not reflect in a better accuracy of the results.

### 2.3. Problem Formulation and ANN Architecture Definition

The problem formulation in the context of supervised learning consists of the definition of the input and output variables and the number of ANN necessary to describe the phenomenon. Different formulations of the problem were analyzed during this research and are referred to herein.

The input variable can be either the pressure–head, or flowrate, data records during a 24–hour period. Since pressure sensors are easier to install, as they do not require any special civil works in pipes (necessary for flow–meters installation), the use of pressure–head data is recommended.

Concerning the output variables, the first approach was to consider the node ID, where the burst occurred, and the burst size. However, the node ID is a random discrete variable with no correlation with the real node coordinates, a huge amount of data being necessary to train the ANN so that it could recognize the burst node location by the node ID number. Thus, the burst location is described by the node coordinates. Thus, two formulations are suggested to describe the burst location: the first is the Euclidian distance to the inlet storage tank, combined with one of its Cartesian coordinates and the second is the use of the node Cartesian coordinates (X, Y).

Concerning the ANN, the problem can be formulated using three independent ANN, one for each output variable, or one unique ANN considering all output variables. The latter is the one recommended herein, since it integrates all the information.

The next step is the ANN architecture definition. A multi–Layer Perceptron (MLP), a class of feedforward artificial neural network (ANN), is used herein. A three–layer ANN is proposed, since the generalization capability of multi–layer perceptron networks is not improved for more than three layers [29,30]. Concerning the number of neurons, several configurations of the ANN should be analyzed and a sensitivity analysis should be carried out to determine the best compromise between ANN simplicity and accuracy of the results. This sensitivity analysis should be carried out using a grid search considering several configurations described as follows: three–layer ANN with multiples of five neurons: (5k + 5, 5k, 5k + 5) being k = 1, . . . , 9.

The Deep Learning Toolbox of MATLAB can be used to configure and to compute the ANN. This specific toolbox uses any ANN in a relatively simple and user–friendly way. The Levenberg–Marquardt (LMA) has been demonstrated to have very good and fast convergence results [31,32], this being the one proposed herein. This algorithm is widely used in optimization problems requiring non–linear least squares curve fitting, with a fast convergence.

### 2.4. ANN Training, Testing and Sensitivity Analyses

Usually, simulated data should be divided for the training and the testing processes. Previous researches [26,27] have demonstrated good results on burst location and sizing in networks when 90% of the datasets are used for training and 10% for testing, these being adopted herein. At this stage, a sensitivity analysis should be carried out to evaluate the number of neurons defined in each layer, the number of datasets to be used and the number of sensors. This is the most time–consuming stage, but also one of the most relevant for successfully locating and quantifying the bursts.

Finally, the selected ANN, among the several analyzed, should be the one that is demonstrated to more successfully predict the burst location, size and time of occurrence. Recommendations should be established for collecting pressure–head (and/or flow–rate) data to continuously use the obtained ANN to detect bursts or anomalous events, based on the real–life measurements.

### 2.5. Application Based on Collected Data

The rationale to apply the proposed ANN approach to real WDS for burst detection is as follows. First, the water utility needs to follow the described methodology to build, train and consolidate the best ANN and to determine the number and location of pressure sensors; this can be carried out by using artificial pressure–head data generated by numerical simulations and by carrying out several sensitivity analysis. At the end of this stage, the trained ANN is ready to be used for burst detection.

Secondly, the pressure sensors need to be installed at the established locations and pressure–data need to be systematically collected and stored by the utility in a centralized data management system. The rationale for leak detection is as follows. The pressure–head data are collected during the 24–hour records and, at the end of the day, it is used to test the ANN. The ANN detects the location and size of anomalous events. The utility analyses and validates obtained ANN results.

This procedure can be applied for near–real time burst location and sizing. This can be carried out by using several ANN trained with shorter time–periods and by running each ANN at every hour of the day. This procedure allows the utility to constantly monitor the network and to detect not only bursts but also sudden changes in demand. The latter can be determined by comparing consumption measurement used for generating the artificial pressure–head and the hourly collected consumption, in that day, at the user level.

Whenever significant consumption changes are observed in the network, for instance, seasonal variation of consumption, or the network operation is changed by closing valves or changing the settings of the pumps, new artificial data need to be generated again for those conditions and the ANN need to be trained with that data.

### 3. Case Study Description

The case study is a water distribution system located in a highly touristic region in the South of Portugal (Quinta do Lago), managed by the water utility InfraQuinta (Figure 2). The WDS supplies approximately 1.7 mm$^3$/year (2018), varying from 2000 to 14,000 inhabitants in winter and summer, respectively, with a relevant seasonality variation in what concerns to water demands. The supplied area has 7.5 km$^2$ and consists mainly of houses/dwellings with swimming pools and irrigated gardens, hotels and some golf courses. Due to the seasonal variation in water demand as well as the increasing scarcity of water in this region year after year, the water utility has installed telemetry at the consumer level, with hourly consumption data for the ca. 2000 consumers, allowing efficient water–use monitoring, especially in the irrigation systems. The network has an approximate length of 77 km with pipes mainly made of asbestos cement and PVC and diameters between 32 and 362 mm. The network hydraulic model has 4448 nodes and 4494 pipes. The network is supplied by one storage tank and four pumping stations located inside the storage tank premises, set to operate at constant pressure during the day. These pumping stations are simulated in the hydraulic model EPANET as constant level storage tanks. Table 1 summarizes the main characteristics of the WDS. Figure 2 presents the analyzed WDS with the location of the existing storage tank and of two consumers: the largest consumer and a small house hold consumer whose daily consumptions are depicted in Figure 3. The largest consumers of the network are hotels and golf courses.

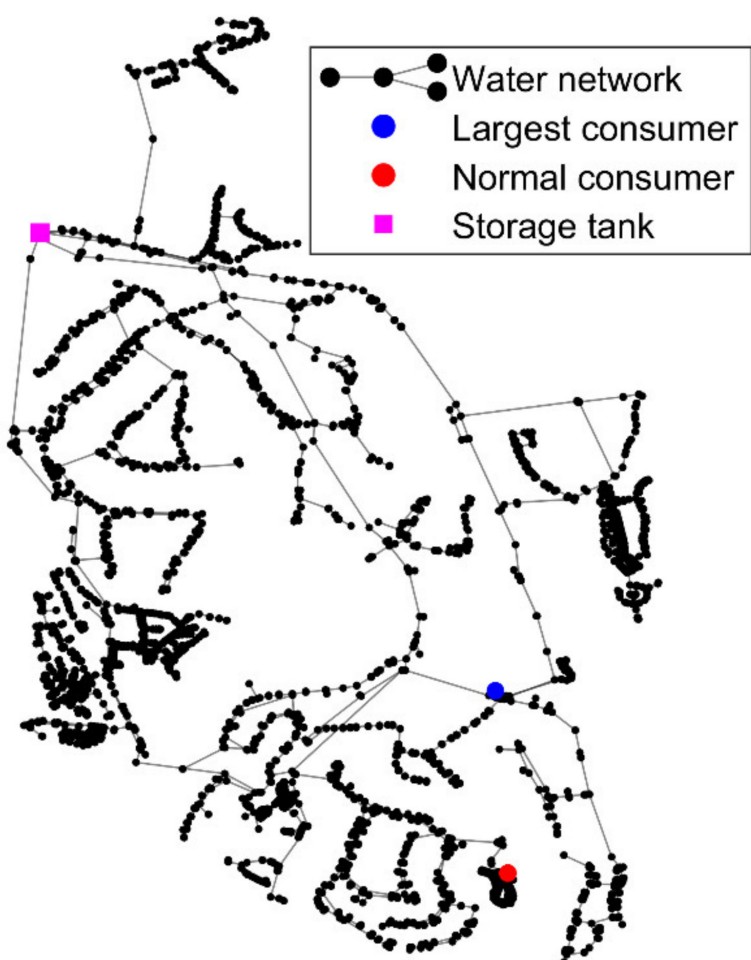

**Figure 2.** Schematic of InfraQuinta network.

**Table 1.** Main characteristics of the WDS.

| | | |
|---|---|---|
| Total Length | | 77 km |
| Area | | 7.5 km$^2$ |
| Inlet of the WDS | | 1 storage tank and 4 pumping stations |
| Average Flowrate at the Inlet | | 70 L/s |
| Pipe Materials | | Asbestos cement and PVC |
| Pipe Diameters | | 32–362 mm |
| Water Supplied Per Year | | 1.7 mm$^3$ |
| Population Supplied | | 2000 (winter) to 14,000 (summer) inhabitants |
| Hydraulic model | Number of pipes | 4494 pipes |
| | Number of nodes | 4448 nodes |

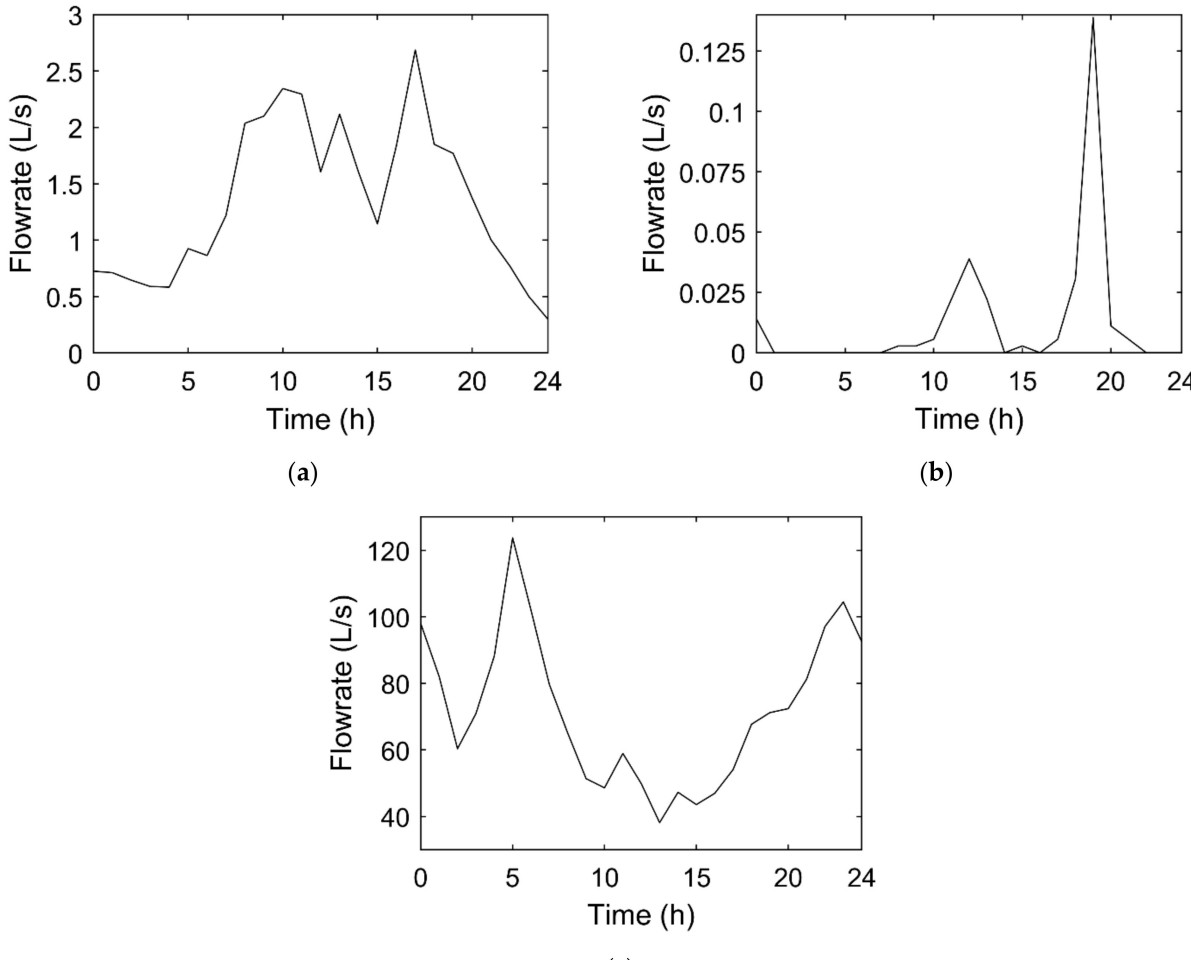

(**a**)

(**b**)

(**c**)

**Figure 3.** Average hourly flowrates for (**a**) the largest consumer, (**b**) a domestic consumer and (**c**) the inlet of the WDS.

The daily flowrate variation of the largest consumer (located in node 8, highlighted in blue in Figure 2) and of a normal household consumer (located in node 1020, highlighted in red in Figure 2) is presented in Figure 3. Major differences are observed in the daily demand patterns of these consumers. The large consumer (a hotel) shows a minimum night consumption, possibly associated with water–night–uses or leakages in the private network, and two consumption peaks during the day, in the morning at 10 a.m. and in the

late afternoon at 6 p.m. (Figure 3a). The individual consumer has also the same two peaks but has hardly any consumption during the remaining time, showing that the users might not be at home during the day (Figure 3b). The average flowrate at the inlet of the WDS is ca. 70 L/s, this being the respective daily variation between 40 and 120 L/s as presented in Figure 3c. A consumption peak at 5 a.m. is probably due to the irrigation of gardens.

## 4. Methodology Application and Sensitivity Analysis

### 4.1. Data Base Construction

The existing seven pressure–head sensors and flowrate meters installed in Quinta do Lago WDS are mainly located near the upstream storage tank, at the inlet of the network, not being widely nor adequately spread throughout the network to allow the pipe burst location based on pressure and flowrate measurements. A detailed analysis of the collected data confirmed that relevant pressure variations were not registered in these sensors, when pipe bursts occurred far from the upstream tank, it not being possible to use collected data during the occurrence of pipe bursts to train the ANN to detect leaks and ruptures This corresponds to the most common situation in real networks.

The solution to the lack of real pipe burst data is to artificially generate pipe burst scenarios using a reliable and calibrated network model. This model can be developed in EPANET, a public domain software applied to WDS modeling, and pipe burst scenarios systematically simulated with the support of any programming tool, such as the MATLAB library (as used in this research). This procedure is applied herein and the artificially generated database is composed of sets of pressure–head data at different locations during a 24–hour period, with a 10–minute interval. Simulated scenarios correspond to six burst sizes, randomly located at different network nodes, starting at different times during the 24–hour period and with a constant 4–hour duration.

The InfraQuinta network model is composed of 4448 nodes. Many of these nodes are very close to each other, corresponding to upstream/downstream nodes of open valves and containing small diameter service connections. Considering all these nodes as potential burst locations would significantly increase the search space and, consequently, the complexity of the burst location and sizing problem. In these cases, search space reduction (SSR) is recommended [19]; this can be carried out based on simple topological analysis. Firstly, every downstream node of the service connection is eliminated from the set of potential burst locations, as its sensitivity is similar to that of the upstream node. This allows a significant reduction of the number of potential burst nodes (from 4448 to 3116 nodes). A second reduction is carried out by removing the downstream node of every valve and nodes with only two connected pipes (except nodes with the service connections). With both simplifications, the number of burst simulated scenarios to train and to test the ANN is reduced to approximately 90%, as the final number of nodes is 276, increasing the efficiency of the search method, as the number of scenarios is significantly lower, with the same expected results.

Six different single burst scenarios are simulated for each of the 276 nodes, described by the emitter law incorporated in EPANET, $Q = CH^{\alpha}$, where C is the emitter coefficient ($m^{3-\alpha}$/h), H is the pressure–head (m) and $\alpha$ is the emitter exponent. Germanopoulos [33] has carried out an extensive study in a real WDN, calibrating the emitter exponent to $\alpha = 1.18$; since then, this value has been widely used by the technical and scientific community [34–36], the reason why it is also adopted herein. Six emitter coefficients (0.05, 0.10, 0.50, 1.0, 1.5 and 2.0) are considered for simulating mean burst sizes between approximately 0.5 and 30 L/s. Each simulated single burst scenario has a constant duration of 4 h and contains the nodal pressure–head and flow rate, with a time step of 10 min, located at one of the 3116 possible locations – these are the nodes obtained after the first SSR. Hence, 18,696 (6 × 3116) different scenarios are simulated, each one with the corresponding pressure–head and flowrate time series, burst starting time, location (represented by the X–Y Cartesian coordinates) and emitter coefficient.

Figure 4 depicts examples of the daily pressure–head time series at two different locations for the scenario corresponding to the burst mean flowrate 25 L/s, located at node 2812, starting at 17 h. Comparing both graphs, Figure 4a shows a clear pressure–head drop at 17 h, whereas, in Figure 4b, the pressure–head hardly shows any variation caused by the burst (the burst effect does not reach the pressure head at the node); this is because the burst occurred closer to the former node (node 1000) than to the latter (node 2000). These graphs highlight the importance of having pressure–head sensors uniformly distributed throughout the WDS, so that bursts located at any node can be captured by, at least, one sensor. The location of these nodes (nodes 1000, 2000 and 2812) is presented in Figure 5a.

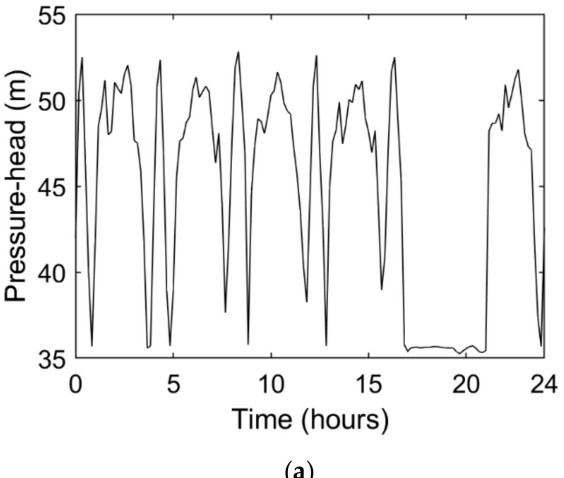

(a)

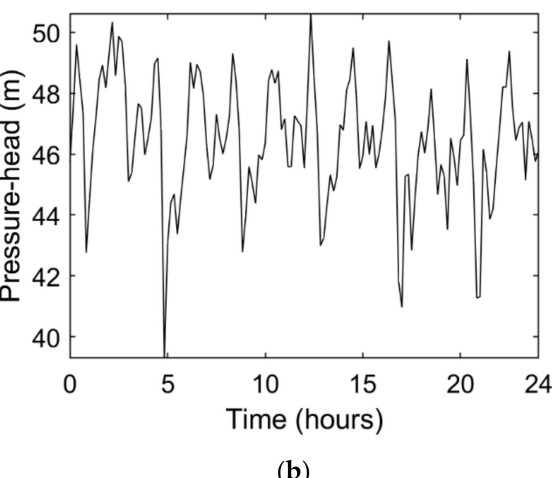

(b)

**Figure 4.** Pressure head series at (**a**) node 1000, and at (**b**) node 2000 in the day of a burst occurrence located at node 2812, at t = 17 h, with Q = 25 L/s.

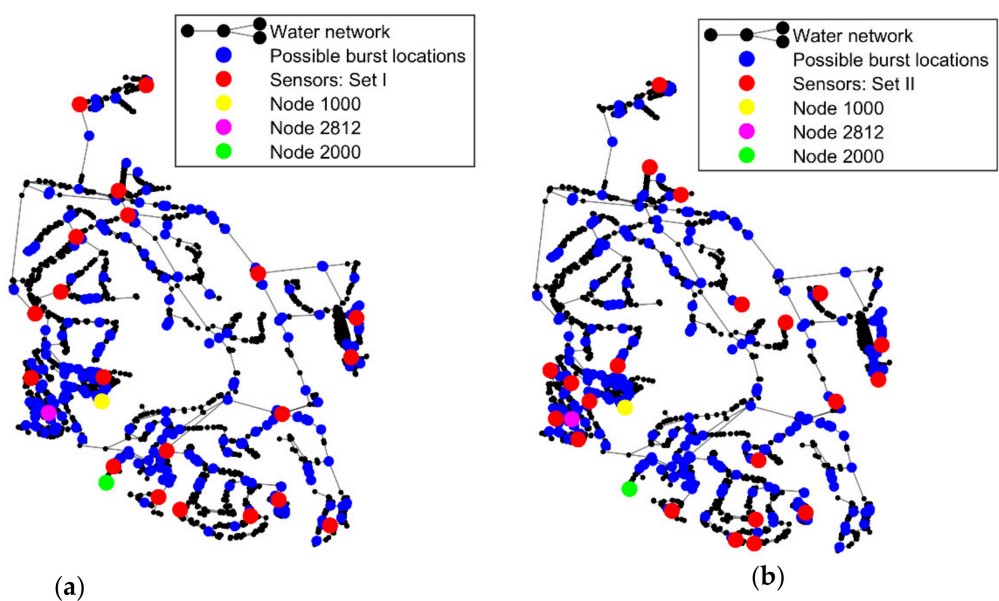

(a)    (b)

**Figure 5.** Water distribution network: possible burst locations and pressure sensors' location: (**a**) Set I – reference location of sensors and (**b**) Set II.

A set of 21 pressure–head sensors (Set I), uniformly distributed throughout the network, is considered as the basis of the analysis. This set corresponds, on average, to one sensor per 3.7 km of pipes, which is considered reasonable for burst detection. The determination of the optimal number and location of pressure sensors is not in the scope of the

current paper. Set I is considered, herein, the reference sensor location. Figure 5a depicts the location of the 21 pressure sensors and of the 276 possible burst locations (after the second SSR).

At a second stage, a sensitivity analysis is carried out to assess the effect of the number of sensors and their location on the success and accuracy of burst location and quantification. Thus, a second set of sensors (Set II) is analyzed; Figure 5b depicts the location of the Set II sensors.

The burst database used to train the ANN is composed of six bursts with 4–hour duration, located at each one of the 276 nodes, leading to a total of 1656 burst scenarios (6 sizes×276 nodes). Each scenario is characterized by 21 records of hourly–determined pressure–head over one day and the characteristics of the bursts, namely the burst location in Cartesian coordinates (X and Y), the burst size, described by the discharge coefficient C, and the burst starting time. Data series with 1 h time step are used herein, instead of 10 min as originally generated by the hydraulic simulator, in order to reduce the ANN training computational time.

### 4.2. Problem Formulation and ANN Architecture Definition

The problem formulation requires the establishment of the input and output variables of the ANN. After preliminary tests in which several combinations of variables of the ANN are analyzed, the configuration with the best results is the one in which the input variables are the pressure–head time series at the 21 sensors corresponding to Set I and the output variables are the node location described by the Cartesian X and Y coordinates and the average burst discharge.

The architecture of an ANN is defined by the input and output data, the number of hidden layers and the number of neurons at each layer. In this paper, the ANN used is of the class Multi–Layer Perceptron (MLP) with three layers, one of which is hidden. Different number of neurons for these three layers are analyzed to determine the configuration that provides the best compromise between lower errors during the training process and better results after testing, along with a reasonable computing time. This process is called herein sensitivity analysis of neurons' number. This analysis considers different numbers of neurons, but maintains the same neuron arrangement of the following type (5k + 5, 5k, 5k + 5) with k ranging from 1 to 9. Figure 6 represents a scheme of the ANN, emphasizing the architecture and the input and output variables.

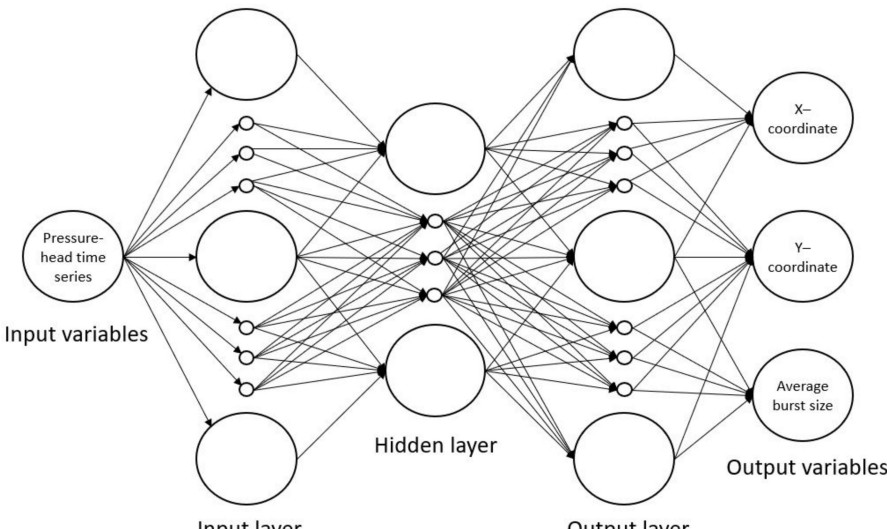

**Figure 6.** Scheme of the Multi–Layer perceptron (MLP) architecture and the input and output variables.

For the ANN development, the burst scenarios are divided in two groups: one with 90% of the scenarios for training the ANN and the other with 10% for testing the ANN. It is guaranteed that every possible leak location had, at least, one of six burst scenarios being used for ANN training. This avoids a given burst location being tested, but not having been trained with any burst, which would significantly decrease the location hit percentage.

The analysis of the ANN configuration is carried out using data from 21 sensors (set I) and 1656 burst scenarios, whose results are presented in Table 2 in terms of the statistical parameters obtained in the training process for three ANN configurations, namely the mean square error (MSE) and the correlation coefficient ($R^2$). These results show that as the number of neurons increases, the MSE diminishes and the $R^2$ increases, highlighting that the more neurons are added, the better the accuracy of the ANN achieved; however, the improvement attained from the ANN with (45, 40, 45) to the (50, 45, 50) are minimum and the computational time significantly increases, the configuration (45, 40, 45) being the one with the best compromise between time and accuracy.

**Table 2.** Main statistical parameters for the training phase for the several ANN configurations (21 sensors with data set I).

| | X–Coordinate | | Y–Coordinate | | Burst Discharge | |
|---|---|---|---|---|---|---|
| Configuration | MSE $(m^2)$ | $R^2$ | MSE $(m^2)$ | $R^2$ | MSE $(l^2/s^2)$ | $R^2$ |
| 40,35,40 | 4.91E04 | 0.95 | 1.76E05 | 0.94 | 7.26 | 0.97 |
| 45,40,50 | 1.52E04 | 0.98 | 6.52E04 | 0.98 | 6.95 | 0.97 |
| 50,45,50 | 1.23E04 | 0.99 | 3.2E04 | 0.98 | 6.34 | 0.98 |

A sensitivity analysis for the number of sensors in the networks is also carried out to assess the effect of the number of sensors in the final results. For this purpose, three ANN are trained with data from different groups of sensors considering Set I: 21 sensors (the reference sensor location), 14 sensors and 7 sensors, each having 1656, 1242 (3/4 of 1656) and 828 (half of 1656) burst scenarios. In terms of the ANN architecture, changing the number of sensors leads to a change in the size of input dataset. Additionally, these results are compared to the results of considering a 2nd set of sensors (Set II) with different locations to emphasize the importance of having a good sensor location uniformly distributed in the WDS. Table 3 presents the statistical parameters, namely the mean square error (MSE) and the correlation coefficient ($R^2$) for the multiple ANN analyzed with the reference configuration (45, 40, 45). These results show that the decrease of the number of sensors and of the burst scenarios used to train the ANN results in the MSE increase and in the $R^2$ decrease, highlighting that the more sensors and the more burst scenarios are used, the better is the obtained accuracy of the ANN. Results from these analysis in the test phase are further discussed in Section 4.3.

*4.3. ANN Training and Testing and Sensitivity Analyses*

4.3.1. Main Results for the Reference ANN

Results presented herein correspond to those obtained for: a three–layer ANN with (45, 40, 45) neurons in each layer, trained and tested with a database composed of 1656 burst scenarios. The input data are composed of hourly pressure–head at the 21 sensors of the Set I (reference set) and the output data are the burst size, described by the burst coefficient C, and the burst location, described by the Cartesian coordinates X and Y. This ANN is considered the reference case and is used for comparison with other results in the sensitivity analysis.

Results from the training phase have been presented in Table 2. Figure 7 presents the results obtained in the testing phase, in terms of (a) the percentage of located pipe bursts regarding the distance uncertainties in X–coordinate and Y–coordinate, (b) the true burst discharge distribution and the respective estimated bursts and burst size relative

uncertainty, given by the ratio between the true and the estimated burst size and the true burst size, (c) the distance uncertainty as a function of the true burst discharge, and (d) burst discharge uncertainty.

**Table 3.** Main statistical parameters for the training phase of the analyzed ANN with the reference configuration (45, 40, 45).

| | | Testing | | | | | |
|---|---|---|---|---|---|---|---|
| | | X–Coordinate | | Y–Coordinate | | Burst Discharge | |
| Sensors (Set I) | Burst Scenarios | MSE (m$^2$) | R$^2$ | MSE (m$^2$) | R$^2$ | MSE (l$^2$/s$^2$) | R$^2$ |
| 21 | | 1.52E04 | 0.98 | 6.52E04 | 0.98 | 6.95 | 0.97 |
| 14 | 1656 | 5.08E04 | 0.94 | 1.51E05 | 0.95 | 7.70 | 0.96 |
| 7 | | 9.53E04 | 0.89 | 1.98E05 | 0.93 | 10.15 | 0.95 |
| 21 | | 1.80E04 | 0.98 | 3.43E04 | 0.99 | 9.83 | 0.95 |
| 14 | 1242 | 2.07E04 | 0.98 | 6.67E04 | 0.98 | 3.63 | 0.98 |
| 7 | | 8.02E04 | 0.91 | 1.40E05 | 0.95 | 10.27 | 0.94 |
| 21 | | 2.05E04 | 0.98 | 2.46E04 | 0.99 | 3.4 | 0.98 |
| 14 | 828 | 4.01E04 | 0.95 | 8.13E04 | 0.98 | 5.52 | 0.97 |
| 7 | | 1.26E05 | 0.86 | 8.65E04 | 0.97 | 9.63 | 0.94 |

Regarding the burst location, the higher the distance uncertainty is, the higher the number of located pipe bursts becomes (Figure 7a). The ANN locates the bursts in 98% of the cases with a maximum uncertainty of 400 m in terms of the X–coordinate and 700 m in the Y–coordinate. However, the ANN can only locate bursts in 60% and 70% of the cases, for the coordinates Y and X, respectively, with uncertainties of 100 m. This 10% difference between the hit percentage on both coordinates results from the fact that the InfraQuinta network is five times longer in the Y–direction than in the X–direction, thus, increasing the search space at the Y–coordinate, decreasing the accuracy of the results.

Concerning the burst size, Figure 7b–d) shows that the highest burst relative uncertainties occur for smaller burst discharges: true burst discharges higher than 15 L/s have size uncertainties lower than 20%, whereas burst sizes below 2.5 L/s have relative uncertainties up to 90–100%. Additionally, for true burst discharges higher than 2.5 L/s, the ANN can successfully predict the burst location with distance uncertainties lower than 250 m (Figure 7c), whereas for lower than 2.5 L/s burst, the distance uncertainty varies between 0 and 700 m. This shows that higher size burst are more effectively located by the ANN than the smaller bursts (as expected).

A sensitivity analysis is carried out, in the following sections, to assess the effect of the ANN configuration in terms of the number of neurons, the number of sensors, the number of burst scenarios used in the training and testing processes and the location of the sensors on the successful burst location and sizing.

### 4.3.2. Effect of the ANN Configuration

Different tested ANN configurations have been analyzed, namely configurations of the type (5k + 5, k, 5k + 5), with k ranging from 1 to 9. The ANN obtained for k = 7, 8, 9 corresponded to those with the highest hit percentage results. The higher the number of neurons in each layer is, the better the results are expected to be, despite increasing considerably the computational time to train the ANN. Thus, a compromise between training time and expected results must be considered.

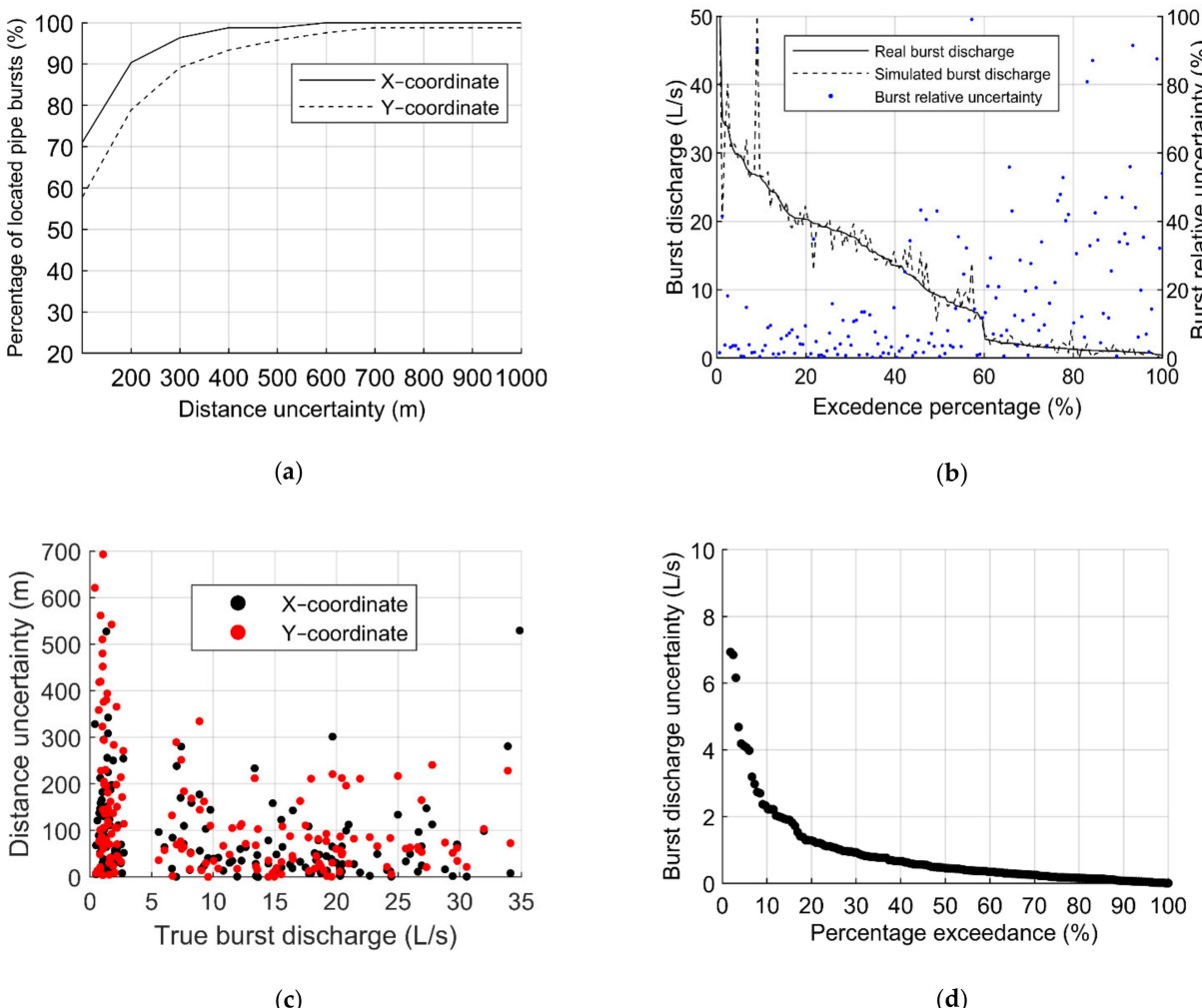

**Figure 7.** Results from the reference ANN: (**a**) percentage of located pipe burst as a function of the coordinates X and Y; (**b**) burst discharge distribution and the respective uncertainty; (**c**) distance uncertainty as a function of the simulated burst discharge; (**d**) burst discharge uncertainties.

Results from the training phase have been presented in Table 2. The ANN configuration (50, 45, 50) presents slightly better results than those obtained for the reference configuration (45, 40, 45); however, it requires excessive time to train (ca. 1.5 days in a Ryzen R9 computed with 32MB RAM), making it impractical to apply in a daily basis in a real WDS. Its results will not be presented herein.

Obtained results from the testing phase of the ANN configuration (40, 35, 40) are compared with those from the reference configuration in Figure 8, in terms of the percentage of located bursts for each distance uncertainty in X and Y directions (i.e., distance between the estimated burst location and the real one). Both ANN have been trained with the same number of burst scenarios (1656) and the same set of sensors (Set I). Figure 8a,b depicts the distance uncertainties of the X and Y–coordinate. Figure 8c represents the percentage exceedance associated with the absolute error in the burst discharge (i.e., the difference between the estimated size of the burst and its real size).

The results for configuration (40, 35, 40) in the prediction of the burst locations (Figure 8a,b) are quite similar to those from configuration (45, 40, 45), with a lower hit percentage of approximately 10% for lower uncertainties. For the minimum distance uncertainty of 100 m, the hit percentage is, on average, 50%, with variations between both coordinates of 15%, being higher in the Y–coordinate. Figure 8c presents similar burst discharge uncertainties for both configurations. Thus, despite the differences between the

configuration (40, 35, 40) and the reference case being minor, the latter (45, 40, 45) provides better overall results.

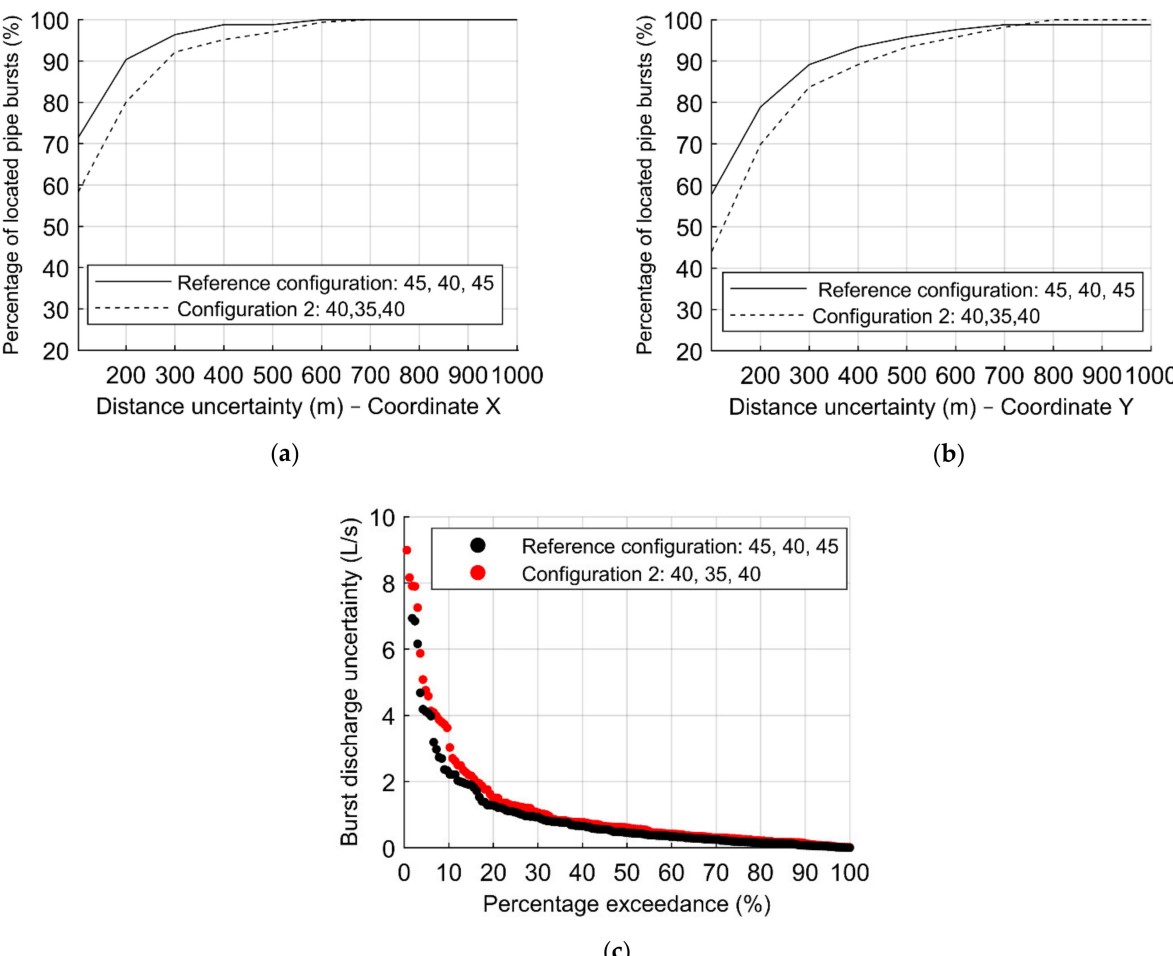

**Figure 8.** Results of the sensitivity analysis to the configuration of the ANN: percentage of located pipe bursts as a function of the coordinate (**a**) X and (**b**) Y; (**c**) burst discharge uncertainties.

### 4.3.3. Effect of the Number of Sensors

The analysis of the effect of the number of sensors on leak location is carried out herein. For this purpose, the Set I of sensors is divided into three groups comprised of 7, 14 and 21 sensors, equally distributed in the network. Once again, the reference configuration is used as a basis to carry out this analysis. Figure 9 presents the location of the three groups of sensors in the WDS.

Results from the training phase have been presented in Table 3. Figure 10 depicts the results from the testing phase, in terms of the percentage of located pipe bursts regarding (a,b) the distance uncertainties in the X–coordinate and Y–coordinate, comparing the hit percentage results concerning the three groups of sensors and (c) the burst discharge uncertainty distribution, for the reference ANN configuration. As observed in the training phase Table 3, there is a noticeable reduction of the percentage of located pipe bursts, especially within the smaller distances (100–300 m), with the decreasing number of sensors considered. The same reduction applies to the burst sizing, as the 21 sensors, considered in the reference case, present fewer burst discharge uncertainties when compared to 14 and 7 sensors. This analysis demonstrates that the higher the number of sensors spread throughout the WDS is, the more successful the ANN can be in the location and quantification of pipe bursts. In the current case, the set with 21 sensors leads to the best location and sizing results.

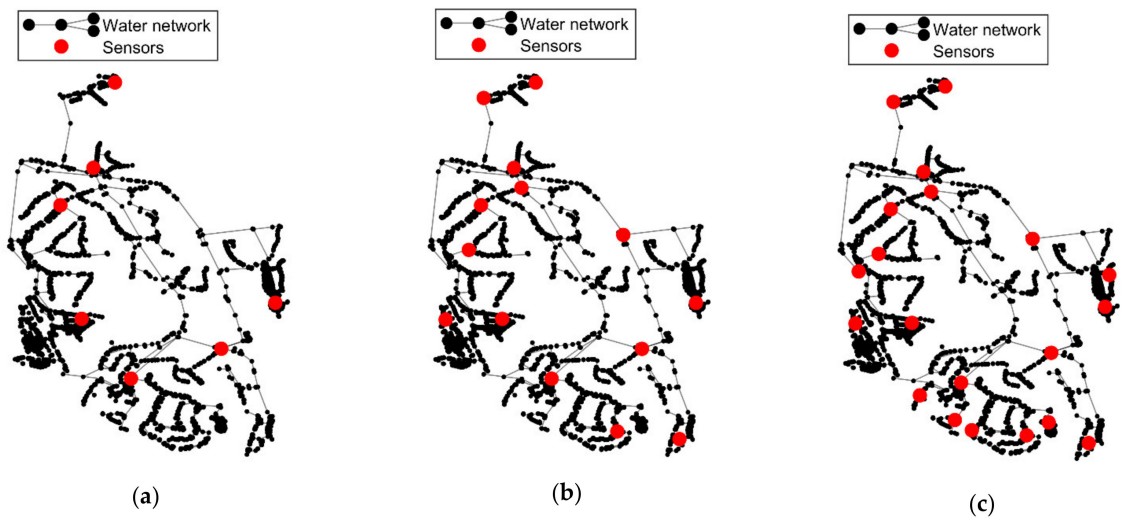

**Figure 9.** InfraQuinta WDS and the 3 groups of sensors composed by (**a**) 7 sensors, (**b**) 14 sensors, and (**c**) the reference group of 21 sensors.

**Figure 10.** Results of the sensitivity analysis to the number of sensors: percentage of located pipe bursts as a function of the coordinates (**a**) X and (**b**) Y; (**c**) burst discharge uncertainties.

### 4.3.4. Effect of the Number of Burst Scenarios Considered

A sensitivity analysis was carried out on the effect of the size of the database, that is the number of burst scenarios considered, for the reference ANN configuration and the number of sensors. For this purpose, three different burst scenarios have been analyzed (828, 1242 and 1656). Results from the training phase have been presented in Table 3 and from the testing phase in Figure 11 in terms of the percentage of located pipe bursts according to the (a–b) distance uncertainty of X and Y coordinates, comparing the hit percentage results concerning the three burst scenarios (828, 1242 and 1656), and the (c) percentage exceedance associated with the absolute error in the burst discharge.

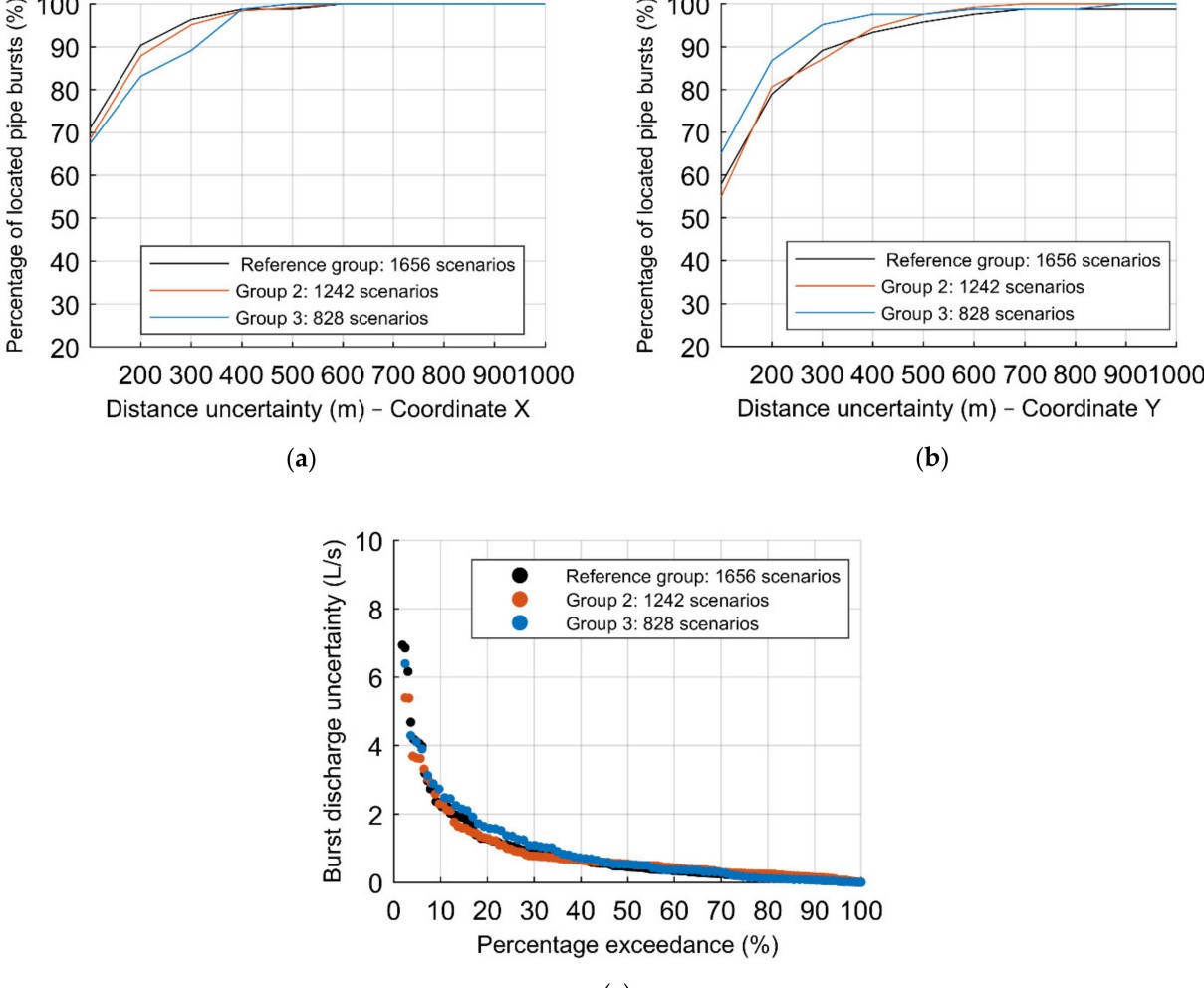

**Figure 11.** Results of the sensitivity analysis to the number of burst scenarios: percentage of located pipe burst as a function of the coordinates (**a**) X and (**b**) Y; (**c**) burst discharge uncertainties.

The ANN trained with 90% of the 828 burst scenarios presents the best results for the location of the pipe bursts concerning the coordinate Y, for the lower distance uncertainty (Figure 11b), contrarily to the results obtained for the coordinate X (Figure 11a), in which it presents the least percentage of located pipe bursts. The reference burst scenario (1656) and 1242 burst scenarios present very similar results with small differences lower than 2% both in terms of pipe location and size.

The results of 828 scenario might seem contradictory, as these present the best results of located pipe bursts on the Y–coordinate. This can be explained by the scenario selection process. Since the scenario location of the bursts is carried out by a random process, selected scenarios turned out to be, by chance, quite well representative along the Y–coordinate.

On the other hand, this is, as expected, the group with the higher uncertainties in locating the burst in the X–coordinate, confirming the good representativity it has along the Y–coordinate. Thus, since the WDS varies less in the X–coordinate than the Y–coordinate, the larger the database is, the better the results become (i.e., lower distance uncertainties are attained).

### 4.3.5. Effect of the Number of Sensors in the Burst Scenarios Considered

After assessing the effect of the number of sensors and of the number of considered scenarios on the burst location and sizing accuracy, it is necessary to carry out a sensitivity analysis on the combined effect of the number of sensors in the different burst scenarios.

Thereby, Figure 12 presents the obtained results from the testing phase on the effect of the number of sensors in the 2nd group of scenarios, comprised of 1242 burst scenarios, with the percentage of located pipe bursts according to the distance uncertainty of the (a) X and (b) Y–coordinate, respectively, comparing the hit percentage results concerning the groups of sensors, and the (c) percentage exceedance associated with the absolute error in the burst discharge. The reference ANN configuration (21 sensors) is used for comparison.

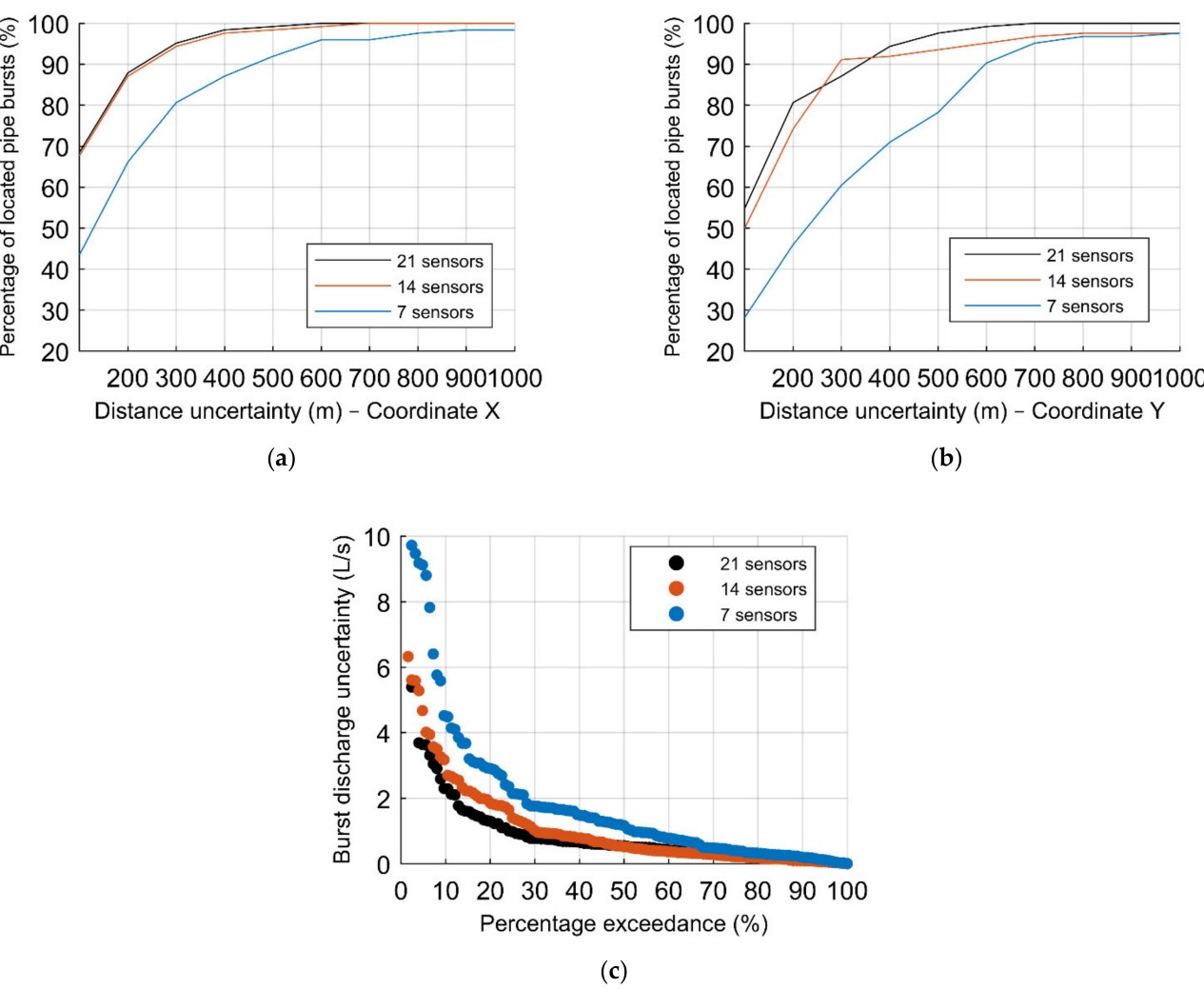

**Figure 12.** Results of the sensitivity analysis for the effect of the number of sensors to 1242 burst scenarios: percentage of located pipe burst as a function of the coordinates (**a**) X and (**b**) Y; (**c**) burst discharge uncertainties.

There is a noticeable reduction regarding the percentage of located pipe bursts, especially within the smaller considered distances (100–300 m) on both coordinates, with the decreasing number of sensors considered (Figure 12a,b). The results between considering

21 and 14 sensors are similar in locating the burst. However, as for quantifying the burst discharge, the ANN trained with data from 21 sensors presents leads to lower size uncertainties than that trained with 14 sensor data (Figure 12c). Considering seven sensors, for the higher precision considered of 100 m, the results are approximately 25% lower than both other groups of sensors.

In addition, Figure 13 depicts the results of the simulations to assess the effect of the number of sensors to the third group of scenarios, composed of 828 burst scenarios, with the percentage of located pipe bursts according to the distance uncertainty of the (a) X and (b) Y–coordinates, respectively, comparing the hit percentage results between the multiple groups of sensors, and the (c) percentage exceedance associated with the absolute error in the burst discharge. The reference ANN configuration is used in all three graphs.

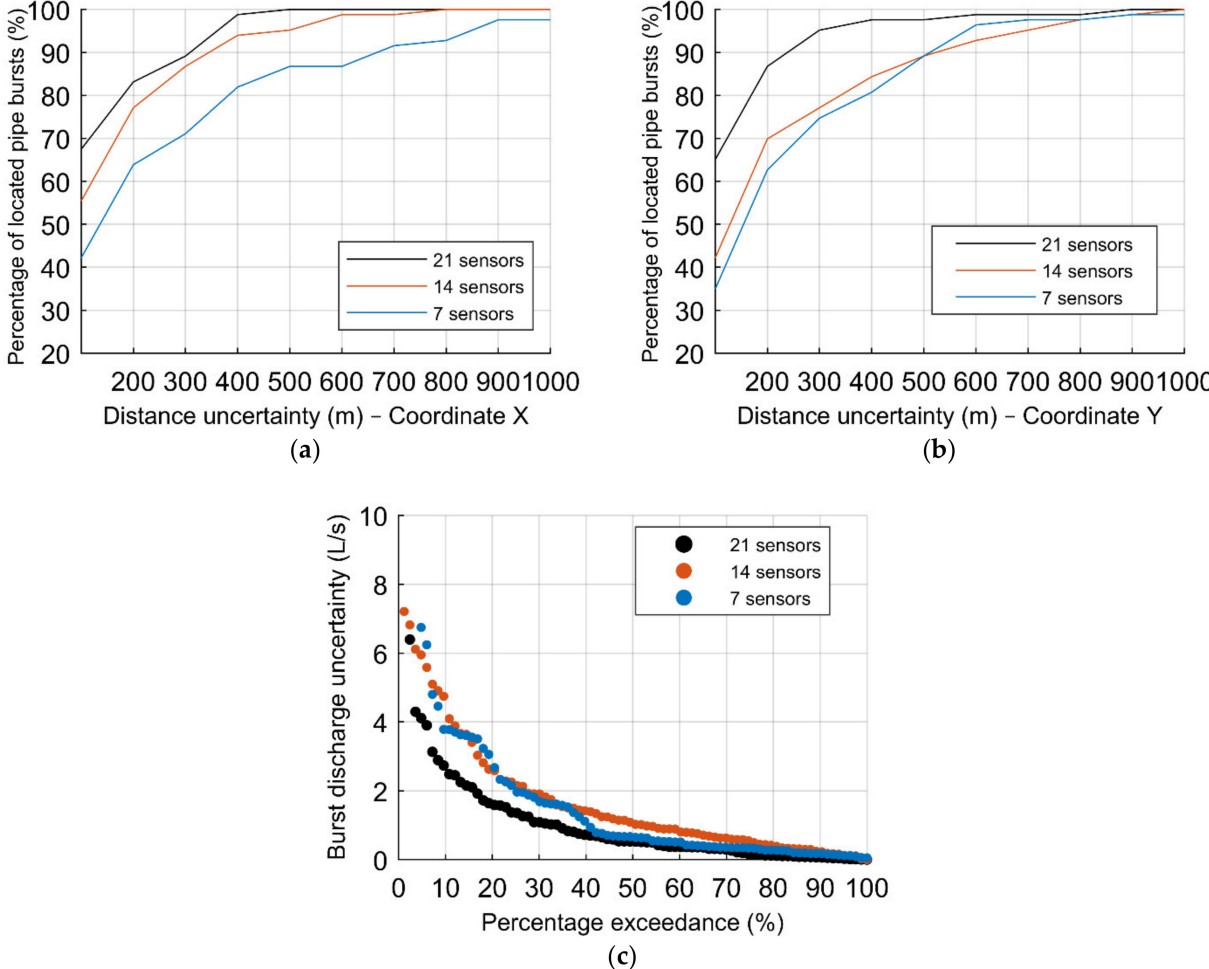

**Figure 13.** Results of the sensitivity analysis for the effect of the number of sensors to 828 burst scenarios: percentage of located pipe burst as a function of the coordinates (**a**) X and (**b**) Y; (**c**) burst discharge uncertainties.

Considering an even smaller database, there is a more evident reduction of well–located pipe bursts for the smaller groups of sensors, especially when considering higher precisions, i.e., smaller distances, for both X and Y coordinates (Figure 13a,b). For the minimum distance of 100 m in the X coordinate, there is a reduction on the percentage of located pipe bursts of 10% and 20%, comparing to the reference group of 21 sensors with 14 and 7 sensors, respectively; considering the Y coordinate, the reduction in the percentage of located pipe bursts is 20% and 30%. The importance of the number of sensors is also visible in quantifying the burst discharge, as the reference group presents the lower burst discharge uncertainties, depicted in Figure 13c.

Thus, smaller databases used to train the ANN need to be compensated with a large number of pressure–head sensors spread throughout the entire WDS to achieve the same results. Overall, the ANN need data to be trained: these data can be provided by fewer burst scenarios but with more measurement locations or by more burst scenarios and fewer sensors.

### 4.3.6. Effect of the Location of Sensors

The effect of the location of the sensors on the ANN successfully detecting bursts is assessed by comparing results obtained for the two sets of sensors, Sets I and II (see the location in Figure 5b). Figure 14 presents the results of both sets of sensors, each comprised of 21 sensors, considering the reference ANN configuration and 1656 burst scenarios, with the percentage of located pipe bursts according to the distance uncertainties of the (a) X–coordinate, and (b) Y–coordinate, and the (c) percentage exceedance associated with the absolute error in the burst discharge. The reference set of sensors, Set I, leads to better results in the location of the pipe bursts in both X and Y coordinates with ca. 5% and 10% higher precision than Set II (see Figure 14a,b). Figure 14c also shows the lower uncertainties considering the Set I, regarding the burst discharge uncertainties. These results are obtained due to the wider spread of the reference set of sensors, throughout the WDS.

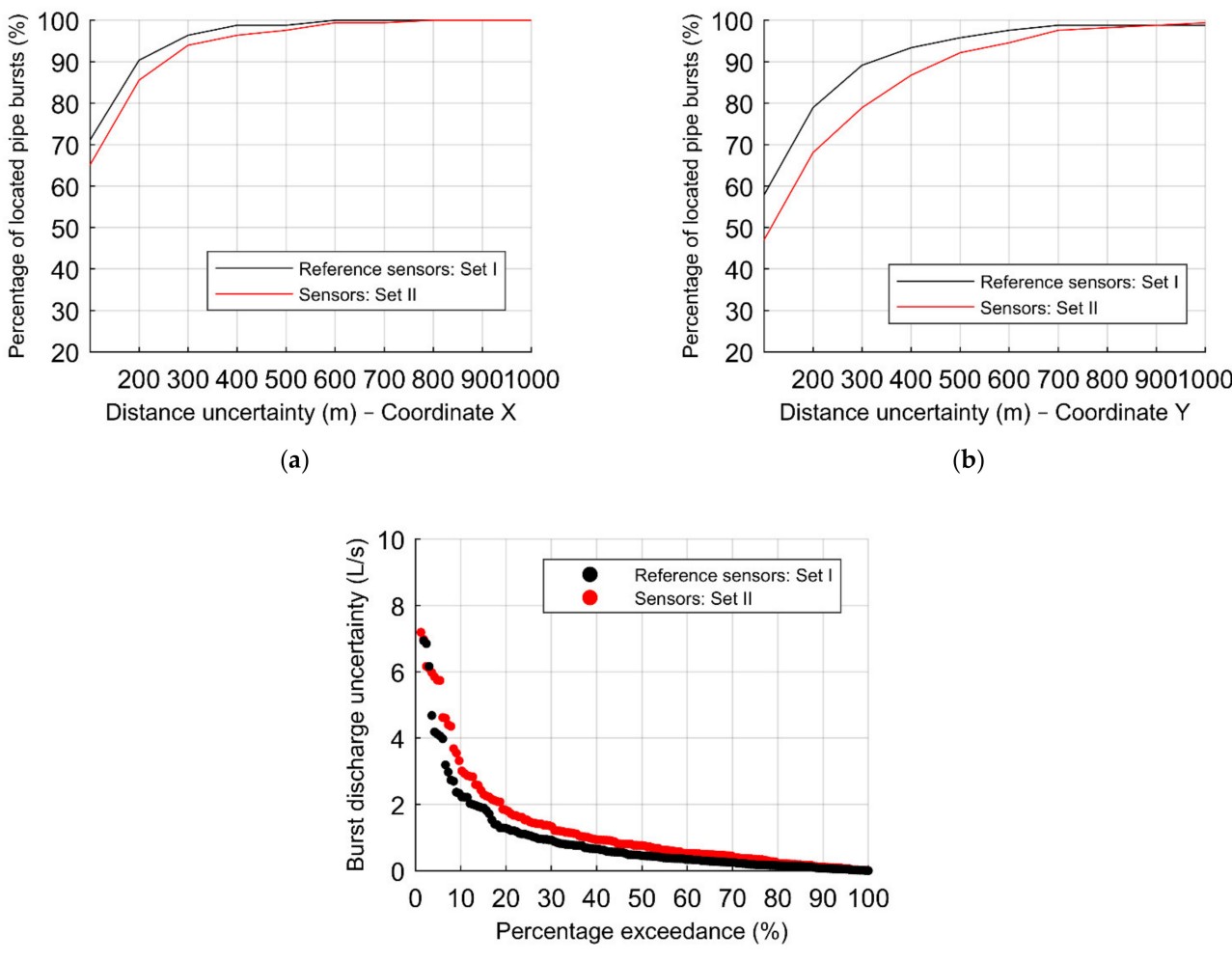

**Figure 14.** Results of the sensitivity analysis for the effect of considering another set of 21 sensors, with the percentage of located pipe bursts as a function of the coordinates (**a**) X and (**b**) Y, and with (**c**) the burst discharge uncertainties.

Additionally, to better assess the effect of considering a different set of sensors, simulations are also carried out for 14 sensors. Figure 15 depicts the percentage of located pipe bursts according to the distance uncertainties of the (a) X–coordinate, and (b) Y–coordinate, as well as the (c) percentage exceedance associated with the absolute error in the burst discharge. See the location of these sets in Figure 16. These results show that, when considering 14 sensors, ANN trained with Set II is more sensible to burst locations. The percentage of located pipe bursts, for Set II, is higher both for the X and the Y–coordinates (see Figure 15a,b). This higher sensitivity to locate the pipe bursts is explained by the location of the 14 sensors; in fact, the 14 sensors of Set II are located at the downstream sections of the WDS (more sensible nodes), whereas the 14 sensors of the reference set are mostly located in intermediate nodes of the water network. Both sets present approximately the same burst discharge uncertainty for 90% of the considered pipe burst scenarios. Figure 16 presents the location of the 14 sensors of both sets, with the reference set (Set I) depicted in red and the Set II in blue.

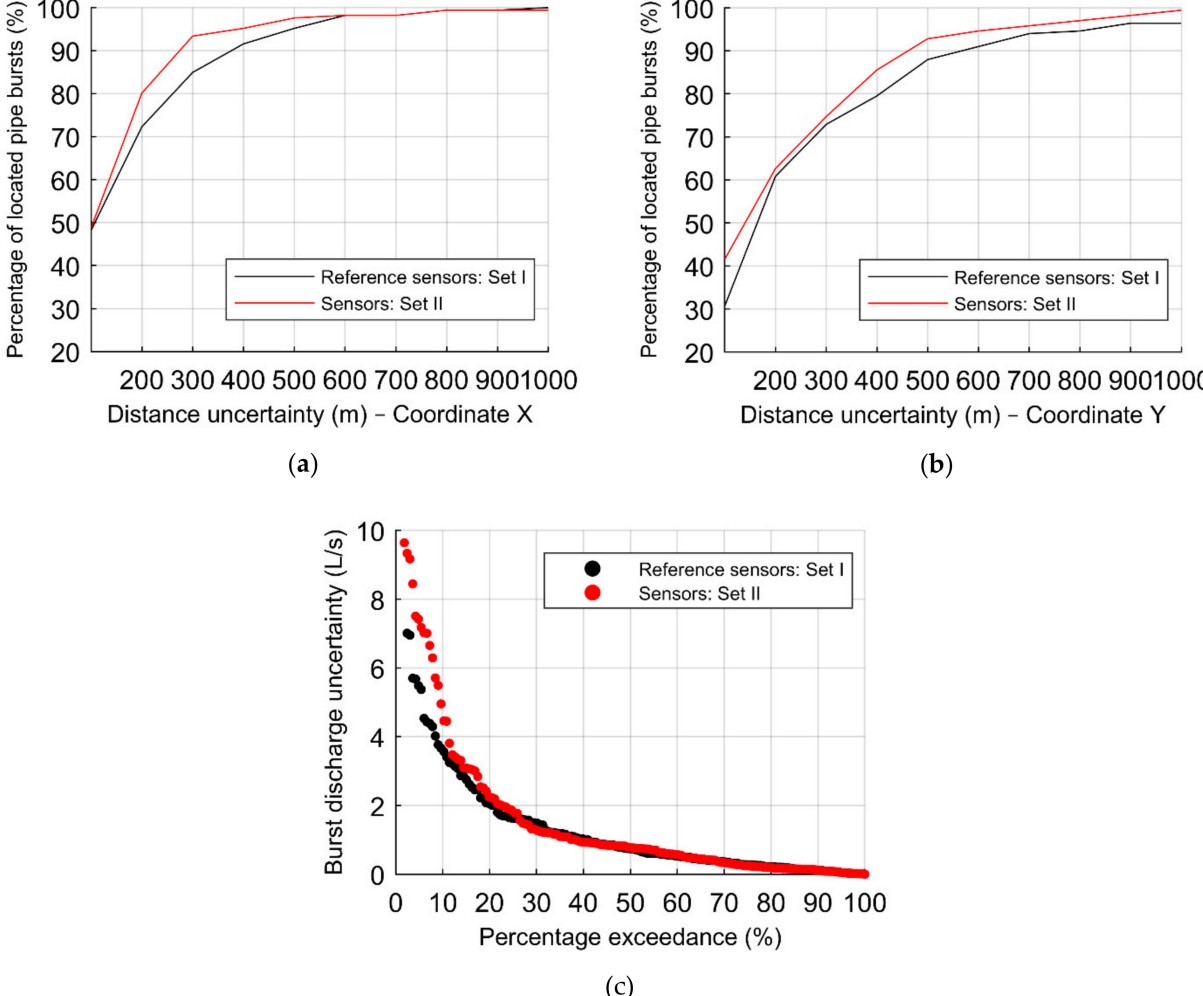

(a)  (b)

(c)

**Figure 15.** Results of the sensitivity analysis to the effect of considering another set of 14 sensors, with the percentage of located pipe bursts as a function of the coordinates (**a**) X and (**b**) Y, and with (**c**) the burst discharge uncertainties.

To conclude the assessment of the effect of considering a different set of sensors, the same analysis is carried out for 7 sensors. Figure 17 depicts the results of both sets, considering the reference configuration and the 1st group of scenarios, composed of 1656 burst scenarios, with the percentage of located pipe bursts according to the distance uncertainties of the (a) X–coordinate, and (b) Y–coordinate, and the (c) percentage exceedance associated with the absolute error in the burst discharge.

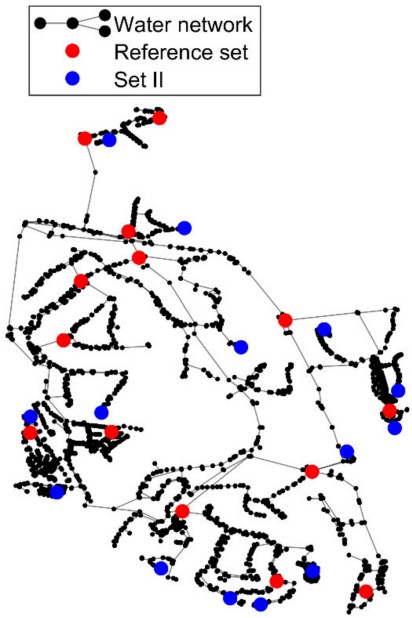

**Figure 16.** WDS with the reference set (Set I) and Set II for 14 sensors.

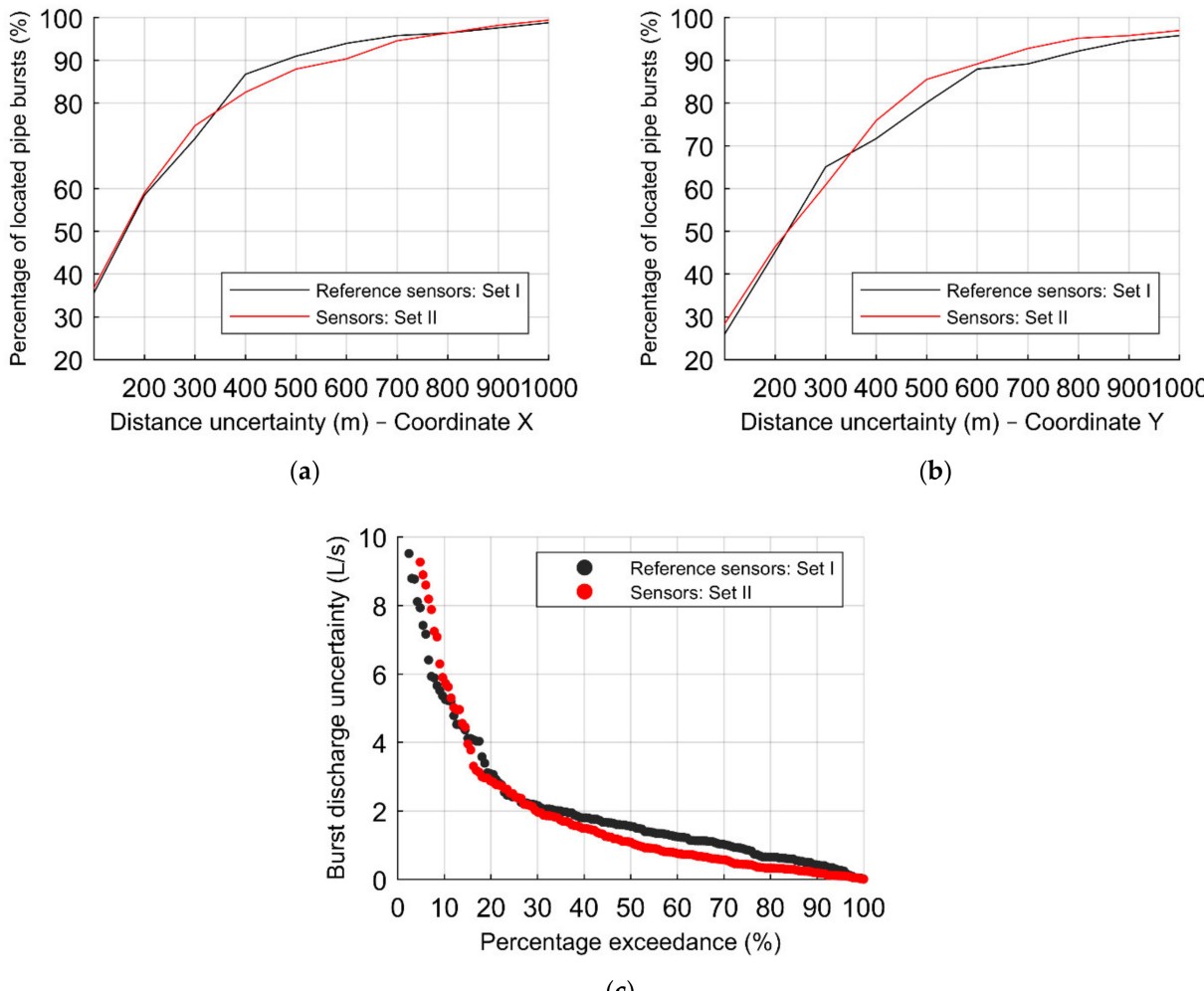

**Figure 17.** Results of the sensitivity analysis for the effect of considering another set of 7 sensors, with the percentage of located pipe bursts as a function of the coordinates (**a**) X and (**b**) Y, and with (**c**) the burst discharge uncertainties.

Both sets present identical percentages of detected pipe bursts according to X and Y coordinates (see Figure 17a,b). Considering 7 sensors, the different location of the sensors seems to become less relevant in comparison with previous cases (14 or 21 sensors), in which there is a clear uniform difference regarding the percentages of located pipe bursts. However, to quantify the size of the burst, the Set II presents better overall results, with lower uncertainties for approximately 85% of the total scenarios. The accuracy of the ANN to estimate the size and location of pipe bursts is highly sensitive to the location (and number) of the sensors. Thus, for future applications, an optimization regarding the location and number of the sensors should be carried out, complementary to this study.

## 5. Conclusions and Further Research

An MLP algorithm was trained and tested to locate and size bursts in a real water distribution network. The ANN allowed us to locate 60–70% of the bursts with an accuracy of 100 m and 98% of the bursts with an accuracy of 500 m. The ANN also can estimate the size of the burst with uncertainties higher than 2 L/s in only 10% of the simulated burst cases and higher than 0.2 L/s in 70% of the cases. A sensitivity analysis of the ANN architecture, number and location of sensors and number of training scenarios has shown that: the ANN with configuration (45, 40, 45) allows the best compromise between accuracy and training time; the higher the number of pressure sensors and the larger the database is, the more successful the burst detection becomes; and the sensors' location significantly affects the success of the burst location, sensors being ideally installed at locations with higher burst detection sensitivity.

Once the best ANN is built, trained, tested and consolidated by using artificial or real data, the ANN is ready to use for burst location and sizing. Then, the pressure sensors need to be installed and pressure–head data collected during the 24–hour used to test the ANN. The ANN detects the location and determines the size of anomalous events. Should this procedure be applied for near–real time burst location and sizing, at every hour of the day, several ANN need to be trained with shorter time–periods and each ANN should be run at every hour of the day, as demonstrated herein for the 24 h. Even if this work uses artificial data generated in hydraulic models, the methodology of leakage localization can be applied to real data acquired by sensors without loss of generality. Nevertheless, real data may have noise or be incomplete, leading to lower accuracy in the results.

The use of ANN has proven to be a very promising machine learning technique for burst detection; however, further research needs to be carried out to be able to apply it in real life systems. Consolidated and well–tested procedures for determining the optimal number and location of pressure sensors need to be developed. The ANN sensitivity needs be tested for different burst durations, sizes and occurring times of the day and to assess the minimum detectable burst size. Different databases with monitoring data and burst–event data need to be integrated in order to be directly used to train the ANN and to more effectively locate pipe bursts.

**Author Contributions:** Conceptualization and methodology, D.C. and L.M.; ANN development and application, M.C. and B.B.; writing—original draft preparation, M.C.; writing—review and editing, B.B., D.C. and L.M.; project management, D.C.; funding acquisition, D.C. All authors have read and agreed to the published version of the manuscript.

**Funding:** This work was supported by the Fundação para a Ciência e a Tecnologia (FCT) through the WISDom project (grant number DSAIPA/DS/0089/2018).

**Institutional Review Board Statement:** Not applicable.

**Informed Consent Statement:** Not applicable.

**Data Availability Statement:** The data presented in this study are available on request from the corresponding author.

**Acknowledgments:** The authors would like to thank the FCT for funding this research through the WISDom project (grant number DSAIPA/DS/0089/2018) and the water utility InfraQuinta for providing data from the case study.

**Conflicts of Interest:** The authors declare no conflict of interest. The funders had no role in the design of the study; in the collection, analyses, or interpretation of data; in the writing of the manuscript, or in the decision to publish the results.

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
