# Peer review of "Near–Real Time Burst Location and Sizing in Water Distribution Systems Using Artificial Neural Networks"

_water, doi:10.3390/w13131841_

Round 1

Reviewer 1 Report

  • It is necessary to specify in the abstract what ANN is, because there are many kinds of ANN.
  •  
  • The author needs to explain why most real and large WDS cannot monitor pressure and discharge using high frequency sensors necessary to measure transient events.
  • In the last sentence (line 67) of the fourth paragraph of the introduction, references should be added to illustrate the feasibility of this study.
  • It is suggested to add a flow chart in Section 2.1 to better illustrate the modeling process.
  • The author used a lot of extent to describe the definition of ANN. I think some of them are unnecessary. I suggest that the author make appropriate deletions.
  • It is recommended to merge sections 2.4 and 2.5.
  • The author should further explain the difference between the data generated by the numerical simulation and the real data, and the impact on the prediction results.
  • Because 7:3 is considered to be the most appropriate ratio in the broad field of machine learning. So the author should explain in the article why the ratio of the training set to the test set is 9:1.
  • The quality of the figures in this article is generally poor, so it is important to redraw the figures.
  • The author should further explain the optimization process of ANN's hyperparameters.
  • The conclusions of this paper are too long, you must concise the conclusions.

Reviewer 2 Report

A novel methodology for near-real time burst location and sizing in water distribution systems (WDS) by means of artificial neural networks (ANN) is carried out in this paper and it is the main subject the authors deal with. They trained an ANN by using pressure-head data generated by a robust and well-calibrated network model (developed in EPANET), that can reliably describe the hydraulic behavior of the system during the 24-h period of analysis and applied it in a case study network in Portugal (Quinta do Lago). In this sense, the authors claim experimental results show that the trained ANN (three-layers with 45-40-45 neurons each) has demonstrated to successfully locate 60-70% of the burst with an accuracy of 100m and 98% of the burst with an accuracy of 500m and to determine burst sizes with uncertainties higher than 2L/s in only 10% of tested cases and higher than 1L/s in 50% of the cases. Authors argue this approach can be used as a daily management tool of water distribution networks. The paper presents quality research on the subject, however, I think that the authors should make an effort to improve the paper by taking into account the following remarks:

  • The neural network design, as well as the different tests carried out, should be better parameterized, for example, in tables for a better presentation, synthesis, and readers' understanding and analysis.

  • A more rigorous statistical validation of the results obtained is necessary to corroborate the reliability of the conclusions. So, it is desirable a clearer experimental design about, to complement the results presented.

Reviewer 3 Report

The paper presents a methodology for the location and sizing of bursts in water distribution networks in nera-real time, using artificial neutral networks. The topic is very interesting and the paper is well written and presented. 

My comments are:

  • In section 4.2 the authors mention that they have performed a sensitivity analysis. However, it is not clear to the readers what are the parameters chosen for the sensitivity analysis, how it is done and what are the results.
  • A table with the basic data for the case study should be presented
  • Secrion 4.3.1.: again the sensitivity analysis is mentioned but there are no data about that and no results are presented. In general in section 4 sensitivity analysis is mentioned several times, but it is not clear what are the parameters examined, how the sensitivity analysis is done and what are the results.
  • Page 11, lines 353-354: three figures are presented but there is no figure caption. The same stands for pages 11-12, lines 356-357, page 12, lines 357-358 and page 13, lines 360-361 and pages 13-14, lines 364-365. It seems that figure 7 is repeated many times?
  •  Please describe the ANN and provide as many data as possible so that this methodology can be used by other researchers.

Round 2

Reviewer 1 Report

Can be published

Reviewer 2 Report

A novel methodology for near-real time burst location and sizing in water distribution systems (WDS) by means of artificial neural networks (ANN) is carried out in this paper and it is the main subject the authors deal with. They trained an ANN by using pressure-head data generated by a robust and well-calibrated network model (developed in EPANET), that can reliably describe the hydraulic behavior of the system during the 24-h period of analysis and applied it in a case study network in Portugal (Quinta do Lago). In this sense, the authors claim experimental results show that the trained ANN (three-layers with 45-40-45 neurons each) has demonstrated to successfully locate 60-70% of the burst with an accuracy of 100m and 98% of the burst with an accuracy of 500m and to determine burst sizes with uncertainties higher than 2L/s in only 10% of tested cases and higher than 1L/s in 50% of the cases. Authors argue this approach can be used as a daily management tool of water distribution networks. The paper presents quality research on the subject. The authors addressed appropriately to the comments and suggestions made in the previous review.